# Polypyrrole-Coated Low-Crystallinity Iron Oxide Grown on Carbon Cloth Enabling Enhanced Electrochemical Supercapacitor Performance

**DOI:** 10.3390/molecules28010434

**Published:** 2023-01-03

**Authors:** Chunhui Wu, Zifan Pei, Menglin Lv, Duchen Huang, Yuan Wang, Shaojun Yuan

**Affiliations:** Low-Carbon Technology & Chemical Reaction Engineering Lab, College of Chemical Engineering, Sichuan University, Chengdu 610065, China

**Keywords:** supercapacitors, anode, low-crystallinity Fe_2_O_3_, polypyrrole, aqueous electrolyte

## Abstract

It is highly attractive to design pseudocapacitive metal oxides as anodes for supercapacitors (SCs). However, as they have poor conductivity and lack active sites, they generally exhibit an unsatisfied capacitance under high current density. Herein, polypyrrole-coated low-crystallinity Fe_2_O_3_ supported on carbon cloth (D-Fe_2_O_3_@PPy/CC) was prepared by chemical reduction and electrodeposition methods. The low-crystallinity Fe_2_O_3_ nanorod achieved using a NaBH_4_ treatment offered more active sites and enhanced the Faradaic reaction in surface or near-surface regions. The construction of a PPy layer gave more charge storage at the Fe_2_O_3_/PPy interface, favoring the limitation of the volume effect derived from Na^+^ transfer in the bulk phase. Consequently, D-Fe_2_O_3_@PPy/CC displayed enhanced capacitance and stability. In 1 M Na_2_SO_4_, it showed a specific capacitance of 615 mF cm^−2^ (640 F g^−1^) at 1 mA cm^−2^ and still retained 79.3% of its initial capacitance at 10 mA cm^−2^ after 5000 cycles. The design of low-crystallinity metal oxides and polymer nanocomposites is expected to be widely applicable for the development of state-of-the-art electrodes, thus opening new avenues for energy storage.

## 1. Introduction

The energy crisis and the environmental pollution problem caused by the excessive use of fossil energy have resulted in higher demand for developing sustainable energy around the world [1,2]. It is highly attractive to develop electrochemical energy storage devices by utilizing renewable energy sources (e.g., tidal, wind, and solar energy) [3,4]. Among the diverse energy storage devices, supercapacitors (SCs) have attracted considerable attention due to their high power density (1–10 kW kg^−1^), fast charge/discharge rate, safety, and long lifespan (>100,000 cycles) [5,6]. Electric double-layer capacitor (EDLC) materials are widely employed in SCs because of their high electrical conductivity, large specific surface area, and low cost [7]. However, the energy storage mechanism of EDLCs occurs at the surface of the electrode, which is responsible for the EDLCs’ poor capacity compared with batteries [8,9]. In addition, according to the equation E = 1/2CV^2^, constructing asymmetric supercapacitor devices (ASCs) can widen the operating voltage window and enhance the energy density [10]. However, the capacitance of ASCs is mainly limited by the inferior capacitance of negative materials.

Metal oxides, such as the pseudocapacitor materials (e.g., Co_3_O_4_, MnO_2_, and Bi_2_O_3_), can offer a higher specific capacitances than EDLC materials because of the charge storage mechanism of surface/near-surface redox reactions [11,12,13,14,15]. Among these metal oxides, Fe_2_O_3_ is one of the most interesting negative materials due to its low cost, environmental benignity, wide operating voltage window (0 to −0.1 V vs. Ag/AgCl), and so on [16,17]. However, it has usually exhibited an insufficient ionic diffusion rate, inferior electronic conductivity, and a volume effect, leading to poor specific capacitance and stability [18,19]. Generally, designing nanomaterials such as nanorods [20], nanosheets [21], and quantum dots [22] can give a sufficient surface area to make good use of active materials. In addition, low-crystallinity nanomaterials are capable of achieving better electrochemical performances than their high-crystallinity counterparts due to their increased structural disorder and defects [23,24]. Xia et al. [19] fabricated a crystalline/amorphous Fe_2_O_3_ nanocomposite to boost the utilization of active materials and enhance the Faradaic reaction, and it delivered a capacitance of 350 mF cm^−2^ at 1 mA cm^−2^. A defective Fe_2_O_3_ nanorod prepared using annealing under an inert atmosphere was reported by Wang et al. [25], and it displayed a capacitance of 381 mF cm^−2^ at 0.5 mA cm^−2^. However, the enhancement of capacitance derived from the boosted valence state change of metal oxide usually causes irreversible conversion in the bulk phase. Therefore, apart from the design of the nanostructure, it is imperative to develop strategies that can transform “bulk” redox reactions into “surface/near-surface” levels of redox reaction, which can simultaneously achieve the capacitance and electrochemical stability.

The construction of nanocomposites by introducing advanced carbon materials or conductive polymers not only favors the electron transfer of electrodes but also hinders the volume effect to improve the cycling stability [26,27]. Polypyrrole (PPy), a heterocyclic conjugate conductive polymer, has attracted considerable attention in energy storage devices [28,29]. Yang et al. [30] prepared a flexible PPy@Fe_2_O_3_@stainless steel yarn composite electrode and enhanced the capacitance compared with the Fe_2_O_3_@stainless steel yarn. Obaidat et al. [31] reposted a PPy-assisted Ag-doping strategy to improve the electrochemical performance of Co(OH)_2_, and the cycling stability was obviously improved after introducing PPy. Compared with the introduction of PPy by chemical oxidation, the electrodeposition strategy is considered to be a promising route for boosting energy storage. PPy prepared using electrochemical deposition methods was reported, and it possessed a molecular ordering formation assisted by an electric field [32]. By depositing a polymer on the surface of electrode to form the core–shell nanostructure, the formative interface between the polymer and the metal oxide can serve as the reservoir to achieve energy storage [33]. Moreover, this interface can efficiently limit ion transfer in the bulk phase, thus hindering the volume effect and further contributing to the electrochemical stability [33]. A low-crystallinity metal oxide with high disorder can usually possess a high charge storage due to the higher active area for enhancing EDLCs and Faradaic pseudocapacitance. However, they usually deliver low cycling stability. A conductive polymer coating layer can serve as the buffer layer to limit the volume expansion of active material and form an active interface to hinder the electrolyte ion transfer in the bulk phase. This could be an efficient strategy to boost the cycling stability. Therefore, we anticipate the construction of low-crystallinity Fe_2_O_3_ coated with a PPy layer, which can simultaneously boost capacitance and electrochemical stability.

In this work, we first constructed low-crystallinity Fe_2_O_3_ nanorod arrays on carbon cloth (D-Fe_2_O_3_/CC) using a simple chemical reduction reaction to enlarge the interface of the electrolyte and electrode. Then, PPy was deposited on the surface of low-crystallinity Fe_2_O_3_ using an electrochemical deposition method to construct a metal oxide/polymer core–shell nanocomposite (D-Fe_2_O_3_@PPy/CC). The NaBH_4_ treatment provided a rougher surface for D-Fe_2_O_3_/CC, favoring the Faradaic reaction derived from the ion diffusion in the bulk phase. The formation of a PPy layer on the surface of D-Fe_2_O_3_ enhanced the charge storage on the surface or near the surface of the electrode. Compared with the pristine Fe_2_O_3_/CC electrode with a capacitance of 114 mF cm^−2^ (123 F g^−1^) at 1 mA cm^−2^, this D-Fe_2_O_3_@PPy/CC electrode delivered an enhanced capacitance of 615 mF cm^−2^ (640 F g^−1^) at 1 mA cm^−2^. Notably, the D-Fe_2_O_3_@PPy/CC electrode also exhibited superior cycling stability, with an initial capacity retention of 79% at a current density of 10 mA cm^−2^ after 5000 cycles, which is higher than the initial capacity retention values of Fe_2_O_3_/CC (42%) and D-Fe_2_O_3_/CC (24%) electrodes.

## 2. Results and Discussion

The scheme of the fabrication of D-Fe_2_O_3_@PPy/CC is displayed in Figure 1a. First, Fe_2_O_3_ nanorod arrays were prepared on carbon cloth using a simple hydrothermal reaction and then immersed in 1 M NaBH_4_ to construct low-crystallinity Fe_2_O_3_ nanoarrays. To investigate the relationship between the reducing process and the electrochemical performance, D-Fe_2_O_3_/CC samples were prepared using various immersion times. After being treated with NaBH_4_, the D-Fe_2_O_3_/CC was coated with PPy to further improve the electrochemical stability. The mount of PPy loading could easily be regulated by changing the electrodeposition time.

Figure 1b displays the Raman spectra of D-Fe_2_O_3_/CC electrodes obtained by changing the reduction time. Clearly, three distinct peaks at 408, 291, and 223 cm^−2^ are observed for the pristine Fe_2_O_3_, D-Fe_2_O_3_-0.5 h, and D-Fe_2_O_3_-1 h, which imply the successful preparation of Fe_2_O_3_ [33,34]. No obvious characteristic peak can be observed in the Raman spectrum of D-Fe_2_O_3_-4 h, which indicates the breakdown of crystallinity on the Fe_2_O_3_ surface by the NaBH_4_ treatment. The Raman spectra of D-Fe_2_O_3_@PPy/CC obtained by varying the electrodeposition time (Figure 1c) also show the characteristic peaks of Fe_2_O_3_, but decreased intensity is observed for D-Fe_2_O_3_@PPy-120 s and D-Fe_2_O_3_@PPy-180 s, which is due to the PPy coating on the surface of Fe_2_O_3_. Notably, the obvious characteristic peaks at 1040, 954, and 916 cm^−1^ are observed, indicating the formation of PPy on the surface of Fe_2_O_3_ [33]. Similar results can be also observed in the XRD patterns of the D-Fe_2_O_3_/CC electrodes. As shown in Figure 1d, the characteristic peaks at 33.0°, 35.7°, 41.06°, 54.4°, and 64.3° are attributed to the (104), (110), (113), (116), and (300) planes of Fe_2_O_3_ (JCPDS#84–0308). The intensity of these characteristic peaks decreased by increasing the reduction time, which indicates the low crystallinity of Fe_2_O_3_ after the NaBH_4_ treatment. The NaBH_4_ treatment led to a reduction in crystal size, thus favoring Na^+^ diffusion near the surface. After being coated with PPy, no obvious change was observed (Figure 1e). This result indicates that the electrodeposition of PPy cannot influence the crystal structure of Fe_2_O_3_.

The surface chemistry of Fe_2_O_3_/CC, D-Fe_2_O_3_/CC, and D-Fe_2_O_3_@PPy/CC were investigated by XPS measurement, as shown in Figure 2. The high-resolution Fe 2p spectrum of Fe_2_O_3_/CC (Figure 2a) exhibits two peaks with binding energies (BEs) of 711.5 and 725.1 eV, corresponding to Fe 2p_3/2_ and Fe 2p_1/2_, respectively. The Fe 2p_3/2_ and Fe 2p_1/2_ peaks with 13.6 eV peak separation are attributed to the typical doublet peaks of Fe^3+^, implying the existence of an Fe^3+^ species [10]. This can be further confirmed by the satellite peak of Fe 2p_3/2_ with BEs at 719.1 eV [17,35]. After the NaBH_4_ treatment, the Fe 2p core-level XPS spectrum of D-Fe_2_O_3_/CC (Figure 2c) exhibited a negative shift compared with that of the pristine Fe_2_O_3_/CC, implying that the NaBH_4_ treatment influences the local chemical states of Fe^3+^. This result indicates the formation of oxygen vacancy on Fe_2_O_3_ [25,36]. The low-crystalline and high-disorder Fe_2_O_3_ can provide more ion diffusion pathways, which further enhance the Faradaic reaction in the bulk phase. The high-resolution O 1s spectrum of Fe_2_O_3_/CC (Figure 2b) shows three peaks. The peaks with BEs at around 530.4, 531.9, and 533.2 eV are assigned to lattice oxygen, chemisorbed oxygen, and physically adsorbed oxygen, respectively [10,37]. Among them, the chemisorbed oxygen, such as O^−^, O^2−^, and O_2_^2−^ on the surfaces of materials, is attributed to the adsorption oxygen on the unpaired electrons generated by the surface defects. The relative contents of these O species can be determined from the proportional areas of the corresponding peaks. After the NaBH_4_ treatment, the O 1s spectrum of D-Fe_2_O_3_/CC (Figure 2d) showed four peaks with BEs at 529.6, 531.5, 533.1, and 535.5 eV, corresponding to the lattice oxygen, chemisorbed oxygen, physically adsorbed oxygen, and water [38,39]. As expected, the content of chemisorbed oxygen presented by D-Fe_2_O_3_/CC was calculated to be 36.8%, which was higher than that of pristine Fe_2_O_3_/CC (27.1%). This further confirmed the higher defect presentation after the chemical reduction procedure. After PPy coating, the wide-scan XPS spectrum of D-Fe_2_O_3_@PPy/CC (Appendix A) exhibited the existence of N, Fe, O, C elements on the surface. The Fe 2p_3/2_ peak at 711.3 eV and the Fe 2p_1/2_ peak at 724.9 eV of D-Fe_2_O_3_@PPy/CC (Figure 2e) showed positive shifts compared with that of D-Fe_2_O_3_/CC. In addition, the high-resolution O 1s spectrum (Figure 2f) displayed two main peaks that can be deconvoluted into four peaks for 530.3, 531.7, 532.8, and 535.7 eV. The content of chemisorbed oxygen was determined to be 28%, which was due to the cover of PPy on the surface of Fe_2_O_3_. Appendix A displays the high-resolution N 1s spectrum, and three deconvoluted peaks are observed at 397.7, 399.8, and 402.1 eV, which are ascribed to the Pyridinic-N, Pyrrolic-N, and nitrogen oxide, respectively [40,41,42]. This further indicates the successful formation of PPy on the surface of Fe_2_O_3_.

The morphology of all electrodes was investigated by SEM measurement. Clearly, the SEM image of Fe_2_O_3_/CC (Figure 3a) displays a nanorod structure, which uniformly covers the CC fiber (Figure 3a inset). Notably, by increasing the NaBH_4_ treatment time the morphology of the nanorod changed. When the reduction time was 0.5 h, some nanosheet structure between the nanorod was observed, as shown in Figure 3b. When the immersion time was more than 1 h, the SEM images of D-Fe_2_O_3_-1 h and D-Fe_2_O_3_-4 h (Figure 3c,d) mainly displayed nanosheet morphology. This could be ascribed to structure conversion by the NaBH_4_ treatment, which provides a strong reduction reaction on the surface of Fe_2_O_3_. A part of an oxygen atom is removed in the reduction reaction, giving a rougher surface for ion transfer.

Figure 4 shows the SEM images of D-Fe_2_O_3_@PPy/CC electrodes prepared using different electrodeposition times. No obvious change could be seen when conducting PPy electrodeposition for 30 s (Figure 4a). As the electrodeposition time increased to 60 s, some nonuniform particles were embedded on the surface of electrode and the nanostructure covered the CC fiber well. When conducting PPy electrodeposition for 120 s, the nanostructure was uniformly coated with PPy (Figure 4c). However, when the electrodeposition increased to 180 s, the nanorod or nanosheet structure could not be found (Figure 4d) and the PPy aggregated on the surface of the electrode (Figure 4d inset). Excessive PPy could obstruct the ion diffusion pathway, thus hindering ion transfer and limiting the utilization of the active material. The TEM image of D-Fe_2_O_3_@PPy (Appendix A) shows the rough surface, which is ascribed to the result of the SEM image. In addition, the selected area electron diffraction (SAED) pattern (Appendix A inset) shows clear diffraction rings, indicating the disordered and polycrystalline features of Fe_2_O_3_.

To investigate the electrochemical performances of all electrodes, a three-electrode system was used in a 1 M Na_2_SO_4_ aqueous electrolyte. Appendix A display the CV curves of D-Fe_2_O_3_/CC electrodes at different scan rates. Clearly, by increasing the NaBH_4_ treatment time, the area of the CV curve increased gradually. The corresponding GCD curves of D-Fe_2_O_3_/CC electrodes are illustrated in Appendix A. The pristine Fe_2_O_3_, D-Fe_2_O_3_-0.5 h, D-Fe_2_O_3_-1 h, and D-Fe_2_O_3_-4 h electrodes delivered capacitances of 113, 701, 960, and 1220 mF cm^−2^ at a current density of 1 mA cm^−2^, respectively. Notably, the capacitance of Fe_2_O_3_/CC can be improved considerably by this chemical reduction treatment. This is ascribed to the formation of a rougher surface, which boosts the ion transfer and the utilization of the active materials. Notably, a weak charge/discharge platform at −0.6 V can be found in the GCD curves for the D-Fe_2_O_3_-0.5 h, D-Fe_2_O_3_-1 h, and D-Fe_2_O_3_-4 h electrodes. This implies enhanced ion transfer and Faradaic reaction in the bulk phase, which could provide additional battery-like charge storage. Figure 5a and Appendix A show the CV curves of D-Fe_2_O_3_@PPy/CC electrodes. All CV curves show relatively quasi-rectangular shapes, implying that they exhibited more surface Faradaic reactions. The corresponding GCD curves (Figure 5b and Appendix A) present relatively asymmetric triangle shapes, further indicating the capacitive behavior of D-Fe_2_O_3_@PPy/CC electrodes. However, as the PPy electrodeposition time increased, the capacitance of the D-Fe_2_O_3_@PPy/CC electrode presented a decreased capacity. This was due to the limitation of Na^+^ transfer in the bulk phase of Fe_2_O_3_ due to the construction of the Fe_2_O_3_/PPy interface, which could favor electrochemical stability during the charge/discharge process. The rate performance of D-Fe_2_O_3_/CC is shown in Figure 5c. Clearly, the pristine Fe_2_O_3_/CC only delivered capacitances of 114, 103, 89, 79, and 69 mF cm^−2^ (123, 111, 96, 85, and 74 F g^−1^) at current densities of 1, 2, 5, 10, and 20 mA cm^−2^, respectively. After PPy coating, the D-Fe_2_O_3_@PPy/CC (D-Fe_2_O_3_@PPy-120 s) electrode (Figure 5d) delivered capacitances of 615, 525, 389, 298, and 208 mF cm^−2^ (640, 547, 405, 310, and 217 F g^−1^) at 1, 2, 5, 10, and 20 mA cm^−2^, respectively. This indicates the enhanced electrochemical performance of the Fe_2_O_3_ electrode using chemical reduction and PPy electrodeposition methods. In fact, the NaBH_4_ treatment can provide more surface roughness for Fe_2_O_3_ and give ion-insertion-like charge storage in the bulk phase, which can enhance the specific capacitance but deteriorate the volume effect. The construction of an Fe_2_O_3_/PPy interface could confine the Na^+^ and limit diffusion in the bulk phase, thus providing more surface Faradaic reactions for charge storage. Appendix A lists the detailed comparisons of Fe_2_O_3_-based and D-Fe_2_O_3_@PPy/CC anodes for supercapacitors [18,19,33,34,43,44,45,46,47,48,49]. The results further indicate the outstanding electrochemical performance of this D-Fe_2_O_3_@PPy/CC electrode.

To further investigate the charge storage mechanism, the surface capacitive behavior and diffusion-controlled contributions were divided using Dunn’s method to analyze the CV data according to the equation i(V) = *k*_1_*v*^1/2^ + *k*^2^*v*, where *k*_1_*v*^1/2^ and *k*_2_*v* represent the contributions from diffusion-controlled (a slow Faradaic reaction) and surface capacitive contributions (including EDLCs and fast surface Faradaic reactions). Usually, the enlarged specific area provided by the nanoarray structure and surface roughness can offer more EDLC capacitance (ion adsorption/desorption). The active sites of the electrode surface can give more Faradaic reactions from the increased chemisorbed redox sites on the surface of electrode. These two contributions compose the surface capacitive behavior. In addition, ion insertion/deinsertion in the near-surface region offers another redox reaction from the diffusion-controlled contribution. Figure 5e compares the diffusion-controlled and capacitive contribution for the pristine Fe_2_O_3_/CC, D-Fe_2_O_3_/CC (D-Fe_2_O_3_-1 h), and D-Fe_2_O_3_@PPy/CC (D-Fe_2_O_3_@PPy-120 s) electrodes. Notably, the diffusion-controlled capacity gradually decreased with the increased scan rate, implying that the entry of Na^+^ into the Fe_2_O_3_ bulk phase was limited at a high scan rate because of the electrochemical kinetic principle [50]. Owing to the “etching” process derived from the NaBH_4_ treatment, Fe_2_O_3_ could react with NaBH_4_ and generate structure defects on the surface of Fe_2_O_3_, offering more active sites and ion diffusion pathways. The diffusion-controlled and capacitive behavior contributions were improved considerably for the D-Fe_2_O_3_/CC electrode. Notably, capacitive contributions of 64%, 69%, 80%, and 87% were achieved for Fe_2_O_3_/CC electrode at scan rates of 10, 20, 60, and 100 mV s^−1^, respectively. After the NaBH_4_ treatment, the D-Fe_2_O_3_/CC electrode delivered decreased capacitive contributions (41%, 46%, 63%, and 76%) at different scan rates. This result confirms the enhanced ion-insertion-like charge storage. After PPy coating, the capacitive contribution further decreased, which was due to the PPy cover on the rough surface of Fe_2_O_3_, thus leading to the limitation of EDLCs. Notably, the electrochemical stability was promoted considerably. As shown in Figure 5f, the D-Fe_2_O_3_@PPy/CC electrode displayed enhanced cycling stability with a capacitance retention of 79.3% at a current density of 10 mA cm^−2^ after 5000 cycles. However, the D-Fe_2_O_3_/CC electrode with a high specific capacitance delivered poor cycling stability, and it only retained 23.9% of its initial capacitance. The pristine Fe_2_O_3_/CC also exhibited an unsatisfied capacitance retention of 42.5%. To further confirm the enhanced stability, the SEM images of Fe_2_O_3_/CC and D-Fe_2_O_3_@PPy/CC after the cycling test are displayed in Appendix A. Clearly, the nanostructure could not be maintained for the Fe_2_O_3_/CC electrode, and the collapse of the nanostructure was observed (Appendix A). Notably, the SEM images of the D-Fe_2_O_3_@PPy/CC electrode show an obvious nanosheet structure (Appendix A), and it covered the CC fiber well (Appendix A inset).

## 3. Materials and Methods

### 3.1. Materials

Polypyrrole (C_4_H_5_N, >99%) was purchased from Aladdin Chemical Co., Ltd. (Shanghai, China). Nitric acid (HNO_3_, >68%), ethanol (C_2_H_5_OH, >99%), ferric chloride (FeCl_3_·6H_2_O, >99%), sodium sulfate (Na_2_SO_4_, >99%), sodium borohydride (NaHB_4_, >99%), and other chemical reagents were purchased from Chengdu Kelong Chemical Reagent Factory, China. The carbon cloth was purchased from Hongshan District, Wuhan Instrument Surgical Instruments (Wuhan, China). All chemical reagents were used as received without further purification.

### 3.2. Synthesis of Fe_2_O_3_/CC

The carbon cloth (CC) was cut into a number of 2 × 3 cm^2^ rectangles, immersed in concentrated HNO_3_, and heated in an oven at 110 °C for 2 h to attach hydroxyl on the surface of CC and remove the impurity on the surface. After the oven cooled to room temperature, the CC was washed with deionized water and ethanol for 30 min under ultrasonic conditions. Subsequently, 0.4 g of FeCl_3_·6H_2_O and 0.24 g of Na_2_SO_4_ were added to 35 mL of deionized water for dissolution. Then, the mixed solution was transferred into a 40 mL Teflon-lined stainless-steel autoclave, and a piece of CC (2 × 3 cm^2^) was immersed in the above solution and treated at 120 °C for 6 h to obtain the FeOOH precursor nanorod on CC. The Fe_2_O_3_ nanorod grown on CC (Fe_2_O_3_/CC) was obtained after heating the precursor at 450 °C in air for 3 h at a heating rate of 2 °C min^−1^. The mass loading of Fe_2_O_3_ on CC was measured to be 0.93 mg cm^−2^.

### 3.3. Synthesis of D-Fe_2_O_3_/CC

The chemical reduction process was conducted using a 1 M NaBH_4_ solution. First, 1.89 g of NaBH_4_ was added to 50 mL of deionized water to form a NaBH_4_ solution. Pieces of Fe_2_O_3_/CC were immersed in the NaBH_4_ solution for different times (0.5, 1, and 4 h) and then washed with water several times. The NaBH_4_ has strong reducibility and can serve as an “activator” to etch the crystallinity of Fe_2_O_3_. By changing the immersion time, the degree of reduction could be adjusted. The obtained low-crystallinity Fe_2_O_3_ on CC was named D-Fe_2_O_3_-xh, where x represents the 0.5, 1, and 4 h reduction times. The mass loading of D-Fe_2_O_3_-1 h was measured to be 0.72 mg cm^−2^.

### 3.4. Synthesis of D-Fe_2_O_3_@PPy/CC

The constant-voltage electrochemical deposition of PPy on low-crystallinity Fe_2_O_3_ on CC was conducted in a standard three-electrode system using D-Fe_2_O_3_/CC, Pt foil, and Ag/AgCl as the work, counter, and reference electrode, respectively. The electrolyte was prepared by adding 0.69 mL of a pyrrole monomer and 2.45 g of NaClO_4_ to 100 mL of deionized water. The mass loading of PPy can be adjusted by changing the electrodeposition time. The applied potential was set to 0.8 V with operation times of 30 s, 60 s, 120 s, and 180 s. Finally, the D-Fe_2_O_3_@PPy/CC electrodes were obtained after washing with deionized water and drying at 60 °C for 8 h. The obtained D-Fe_2_O_3_@PPy/CC electrodes with different electrodeposition times were named D-Fe_2_O_3_@PPy-30 s, D-Fe_2_O_3_@PPy-60 s, D-Fe_2_O_3_@PPy-120 s, and D-Fe_2_O_3_@PPy-180 s. The mass loading of D-Fe_2_O_3_@PPy-120 s was measured to be 0.96 mg cm^−2^.

### 3.5. Material Characterization

A Raman microscope (λ = 455 nm; Thermo Fisher Scientific Inc., Waltham, MA, USA), was used to obtain the Raman dates of all samples. A Philips PW1730 X-ray diffractometer was used to collect X-ray diffraction (XRD) patterns. The X-ray photoelectron spectroscopy (XPS) spectra of all samples were collected using an X-ray photoelectron spectrometer (AXIS NOVA, Kratos, UK). The scanning electron microscopy (SEM) images were characterized by an emission scanning electron microscope (Regulus 8230 field, Hitachi, Tokyo, Japan).

### 3.6. Electrochemical Measurement

A CHI660E electrochemical workstation was used to evaluate the electrochemical performance, including cyclic voltammetry (CV) and galvanostatic charge–discharge (GCD). All electrodes were measured in a three-electrode system using Pt foil as the counter electrode, Ag/AgCl as the reference electrode, and 1 M Na_2_SO_4_ as the electrolyte. All electrochemical performances were tested at 25 °C and ambient pressure. The specific capacitance (C, mF cm^−2^) was collected by the GCD and CV data according to the equations *C* = *I* × Δt/(*S* × Δ*V*) and *C* = ∫IdV/(2 × Δ*V* × *v* × *S*), respectively, where *I* (mA), ∫IdV, ΔV (V), Δt (s), and v (V s^−1^) represent the discharging current, the area of the CV curve, the potential window, the discharge time, and the scan rate, respectively. *S* (cm^2^) represents the geometric area of the working electrode (in this work, *S* = 0.5 cm^2^).

## 4. Conclusions

In summary, we demonstrated polypyrrole-coated low-crystallinity Fe_2_O_3_ supported on carbon cloth as the self-support anode for SCs. The chemical reduction treatment offered more surface roughness for D-Fe_2_O_3_/CC, favoring the Faradaic reaction derived from the ion diffusion in the bulk phase. The construction of a Fe_2_O_3_/PPy interface further enhanced the charge storage on the surface or near the surface of the electrode, thus efficiently hindering the volume effect of Fe_2_O_3_ during the charge/discharge process. Compared with the pristine Fe_2_O_3_/CC electrode, the enlarged surface roughness and Fe_2_O_3_/PPy interface gave more active sites for both Faradaic reactions and ion transfer, and the capacity contributions from diffusion-controlled and capacitive behaviors were simultaneously improved. The D-Fe_2_O_3_@PPy/CC electrode showed a specific capacitance of 615 mF cm^−2^ at 1 mA cm^−2^ and still retained 79.3% of its initial capacitance at 10 mA cm^−2^ after 5000 cycles.

## Figures and Tables

**Figure 1 molecules-28-00434-f001:**
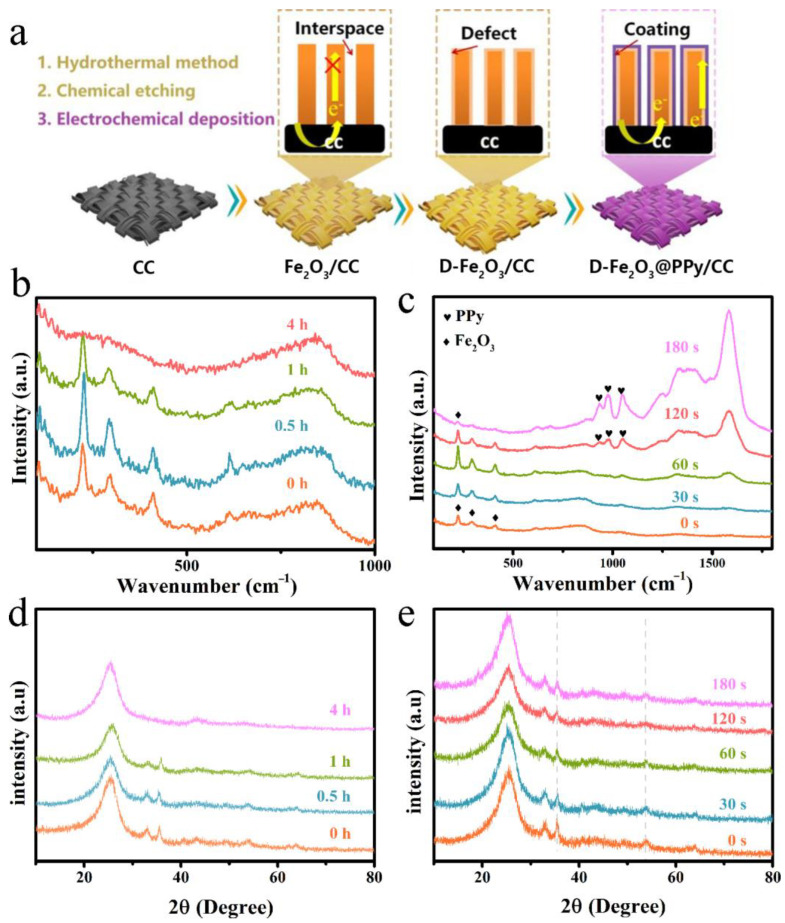
(**a**) Schematic illustration of the fabrication process of a D-Fe_2_O_3_@PPy/CC electrode. Raman spectra of (**b**) D-Fe_2_O_3_-xh (x = 0.5, 1, and 4) and (**c**) D-Fe_2_O_3_@PPy-ys (y = 30, 60, 120, and 180) electrodes. The corresponding XRD patterns of (**d**) D-Fe_2_O_3_-xh and (**e**) D-Fe_2_O_3_@PPy-ys electrodes.

**Figure 2 molecules-28-00434-f002:**
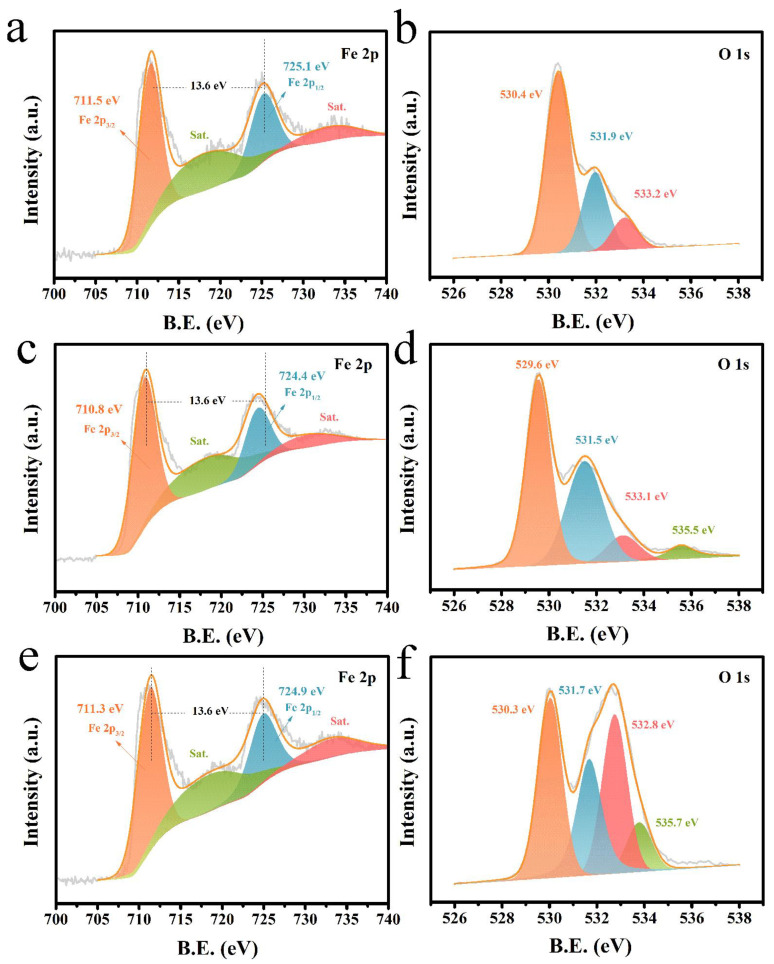
The XPS spectra of Fe_2_O_3_/CC for (**a**) Fe 2p and (**b**) O 1s regions. The XPS spectra of D-Fe_2_O_3_/CC for (**c**) Fe 2p and (**d**) O 1s regions. The XPS spectra of D-Fe_2_O_3_@PPy/CC for (**e**) Fe 2p and (**f**) O 1s regions.

**Figure 3 molecules-28-00434-f003:**
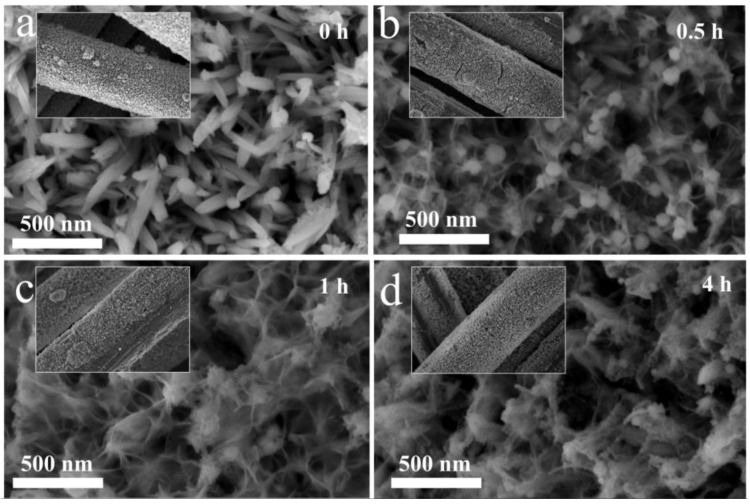
SEM images of D-Fe_2_O_3_/CC electrode prepared using different immersion times: (**a**) pristine Fe_2_O_3_, (**b**) D-Fe_2_O_3_-0.5 h, (**c**) D-Fe_2_O_3_-1 h, and (**d**) D-Fe_2_O_3_-4 h.

**Figure 4 molecules-28-00434-f004:**
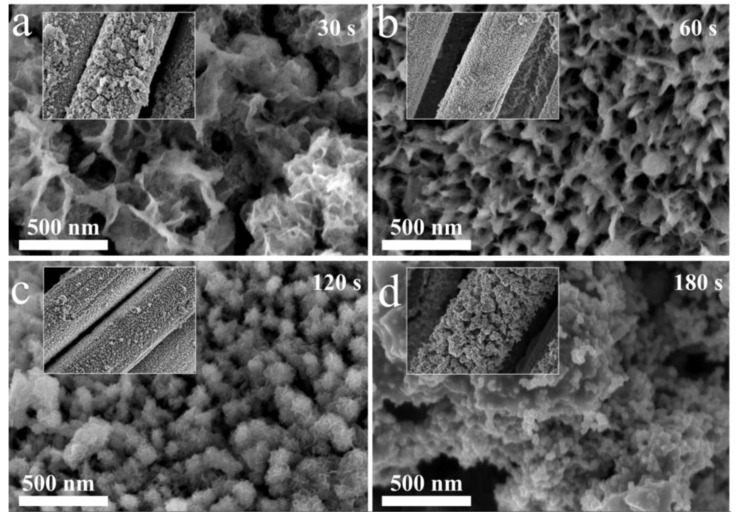
SEM images of D-Fe_2_O_3_@PPy/CC electrode prepared using different electrodeposition times: (**a**) D-Fe_2_O_3_@PPy-30 s, (**b**) D-Fe_2_O_3_@PPy-60 s, (**c**) D-Fe_2_O_3_@PPy-120 s, and (**d**) D-Fe_2_O_3_@PPy-180 s.

**Figure 5 molecules-28-00434-f005:**
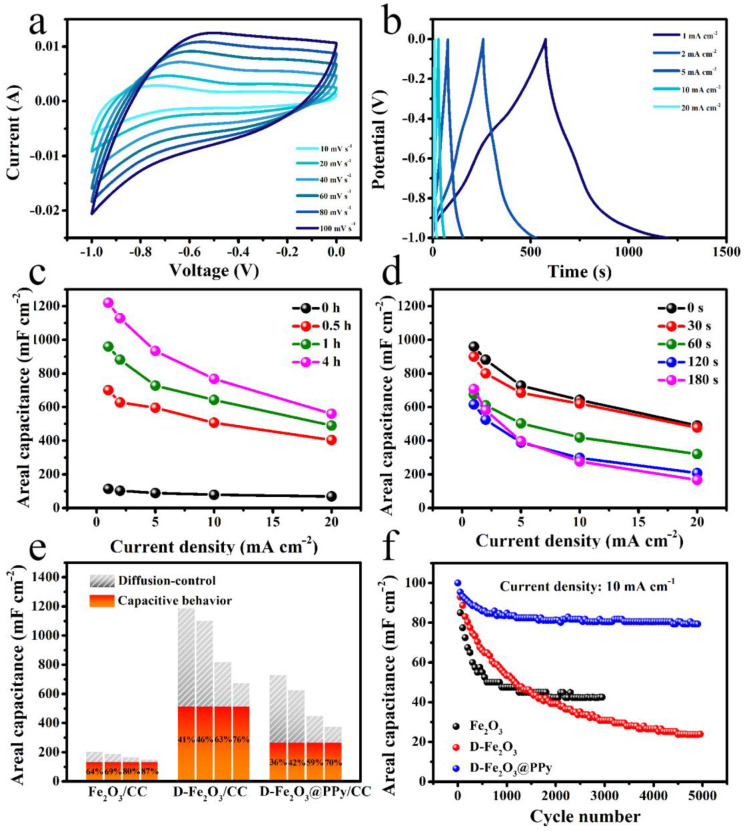
(**a**) CV curves of D-Fe_2_O_3_@PPy/CC (D-Fe_2_O_3_@PPy-120 s) at different scan rates. (**b**) GCD curves of D-Fe_2_O_3_@PPy/CC (D-Fe_2_O_3_@PPy-120 s) at different current densities. (**c**) Rate performances of D-Fe_2_O_3_/CC electrodes. (**d**) Rate performances of D-Fe_2_O_3_@PPy/CC electrodes. (**e**) Diffusion and capacitive contributions of Fe_2_O_3_/CC, D-Fe_2_O_3_/CC (D-Fe_2_O_3_-1 h), and D-Fe_2_O_3_@PPy/CC (D-Fe_2_O_3_@PPy-120 s) electrodes at scan rates of 10, 20, 60, 100 mV s^−1^. (**f**) Long-cycling performances at current density of 10 mA cm^−1^ for 5000 cycles.

## Data Availability

The data are contained within the article.

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
