# Peer review of "Polypyrrole-Coated Low-Crystallinity Iron Oxide Grown on Carbon Cloth Enabling Enhanced Electrochemical Supercapacitor Performance"

_molecules, 2023, doi:10.3390/molecules28010434_

Round 1
Reviewer 1 Report
Thank you so much for such a great work on the testing of cloth based supercapacitor performance.
Comments:
1. You did not mention the testing thermodynamic conditions, I assume these are standard conditions.
Line 5: and is needed between Yuan Wang,* and Shaojun Yuan*
Lines 297-298: Figure 5 e, the percentage numbers will be hard to read.
Line 309: There is a typo "...in bulk phase. the construction ..."
This topic is not original but relevant to the field and can be considered as a good addition to the scientific research work. There are many research works in this field. Some of them are mainly focused on the iron oxide nanoparticles (10.1038/ncomms14264 ;
https://doi.org/10.1021/acsomega.9b03869 ; and etc) to improve capacitance and cyclic stability. Some of them exercise a new design using polypyrrole as well ( https://doi.org/10.1021/acsaem.8b02167; https://doi.org/10.3390/met11060905) to improve the capacitance and cyclic stability. Systematic investigations are still going on in this field with some success, trying different combinations of electrode materials creating novel advanced nanocomposites, metal-organic frameworks, and other hybrid structures.
I would only suggest reporting the thermodynamic conditions at which the electrochemical testing has been performed. I might be missing just that.
Figure 5 (e) shows higher diffusion-control and capacitance contribution for D-Fe2O3/CC and not for D-Fe2O3/PPy/CC electrode at different scan rates, but claiming a better cycling performance for D-Fe2O3/PPy/CC electrode according to Figure 5 (f). It is not clear to me why would it be like that. Should D-Fe2O3/PPy/CC electrode have more capacitance and diffusion controlled processed since it is covered with PPy as it is mentioned in conclusion section?
In Figure 5 (c) and 5 (d) the actual letters for those figures are crossing the parenthesis. As I mentioned before, the Figure 5 (e) should be improved, the percentages cannot be read by a reader, there is a typo in conclusion ( line 309).
Author Response
Reviewer 1:
- You did not mention the testing thermodynamic conditions, I assume these are standard conditions.
Response:
We appreciate for the reviewer’s good suggestion on revising our manuscript. The manuscript has been carefully revised following the reviewer’s suggestion accordingly. As suggested, we added the related description in experiment section (Please see line 152–153).
- Line 5: and is needed between Yuan Wang,* andShaojun Yuan*
Lines 297-298: Figure 5 e, the percentage numbers will be hard to read.
Line 309: There is a typo "...in bulk phase. the construction ..."
Response:
As suggested, the spelling mistakes and typos in the article were corrected accordingly.
- This topic is not original but relevant to the field and can be considered as a good addition to the scientific research work. There are many research works in this field. Some of them are mainly focused on the iron oxide nanoparticles (10.1038/ncomms14264; https://doi.org/10.1021/acsomega.9b03869; and etc) to improve capacitance and cyclic stability. Some of them exercise a new design using polypyrrole as well ( https://doi.org/10.1021/acsaem.8b02167; https://doi.org/10.3390/met11060905) to improve the capacitance and cyclic stability. Systematic investigations are still going on in this field with some success, trying different combinations of electrode materials creating novel advanced nanocomposites, metal-organic frameworks, and other hybrid structures.
Response:
Previous works have demonstrated that the low-crystallinity Fe2O3 or metal oxide exhibited a better capacitance than the high-crystalline counterpart because of their more structural defects and disorder. As far as we know, it is still a tremendous challenge to obtain anode materials with high capacitance, good rate capability and excellent cycling stability. Therefore, we anticipate a low-crystallinity Fe2O3 coated by conductive polymer layer that can simultaneously boost capacitance and electrochemical stability. We further updated the description of introduction to describe the novelty and importance of the work, and cited the related references (Please see line 65–71, line 77–84).
- I would only suggest reporting the thermodynamic conditions at which the electrochemical testing has been performed. I might be missing just that.
Response:
As suggested, we added the related description in experiment section.
- Figure 5 (e) shows higher diffusion-control and capacitance contribution for D-Fe2O3/CC and not for D-Fe2O3/PPy/CC electrode at different scan rates, but claiming a better cycling performance for D-Fe2O3/PPy/CC electrode according to Figure 5 (f). It is not clear to me why would it be like that. Should D-Fe2O3/PPy/CC electrode have more capacitance and diffusion controlled processed since it is covered with PPy as it is mentioned in conclusion section?
Response:
D-Fe2O3@PPy/CC electrode displayed a relatively low capacitance and higher cycling stability, which is an interesting found. In fact, the NaBH4 treatment provide more defect for Fe2O3, which can enhance the capacitance especially the faradaic reaction in the bulk phase or near surface of Fe2O3. However, the boosted faradaic reaction in bulk phase can lead to the volume effect during the charge/discharge process. When constructing PPy layer on the surface of D-Fe2O3, Na+ can be fixed on the interface of Fe2O3/PPy favorably, and the Na+ diffusion in bulk phase can be limited, thus efficiently hinder the volume effect. It also resulted in the reduction of faradaic pseudocapacitance derived from the diffusion-controlled contribution. In addition, the PPy coating covered the roughness surface of Fe2O3, which also resulted in a decreased EDLC. Therefore, D-Fe2O3/PPy/CC electrode have decreased capacitive and diffusion-controlled contributions compared with D-Fe2O3/CC electrode. Notably, the modified D-Fe2O3@PPy/CC delivered a superior electrochemical performance compared with pristine Fe2O3/CC, which indicates the efficient strategy for enhancing electrochemical supercapacitor performance.
- In Figure 5 (c) and 5 (d) the actual letters for those figures are crossing the parenthesis. As I mentioned before, the Figure 5 (e) should be improved, the percentages cannot be read by a reader, there is a typo in conclusion (line 309).
Response:
As suggested, we updated the Figure 5c, 5d, and 5e accordingly. The typo was corrected.

Reviewer 2 Report
Review report:
Authors reported “Polypyrrole-coated low-crystallinity iron oxide grown on carbon cloth enabling enhanced electrochemical supercapacitor performance”. The organization of this work is good, and the discussion is well organized. The characterization and calculation are both solid for the conclusion. Nevertheless, I have some comments which are listed below.
1. The synthesis scheme is not clear, and it should be revised in a detailed way.
2. The calculation formula of specific capacity ( mF cm-2) is not accepted by researchers. Specific capacity and specific capacitance are two concepts. The author shall unify the units as F g-1 or mA h g-1.
3. The author claimed that their work is a novel investigation, however, myriads of works related to “ D-Fe2O3@PPy “ have been published to date. So, authors should change the way of the presentation focusing on novelty. The introduction should be improved with a paragraph describing the novelty and importance of the work.
4. The authors must carefully claim their novelty in the INTRODUCTION. In addition, the authors need to do some formatting errors that should be carefully checked and corrected in the text.
5. The source and purity of all chemicals used should be specified.
6. The surface area measurements are very important for active electrodes in supercapacitor applications. Please provide nitrogen adsorption and desorption (BET analysis) of the active materials (D-Fe2O3@PPy).
7. The authors have missed their mass loadings of “1.) MnS; 2.) Ni(OH)2, and 3.) MnS@Ni(OH)2” electrode materials should be included in the revised manuscript.
8. The selected area electron diffraction (SAED) is very crucial in supercapacitor applications. Please provide SAED of the active materials (D-Fe2O3@PPy).
9. The authors should confirm that in the “Experimental section” in the Electrochemical studies having a “ 1 M Na2SO4” electrolyte solution. Here, my point is why the author not used ‘‘Li2SO4, SPVA-HRG, 1 M H2SO4’ and ‘other sources’ instead of ‘Na2SO4’ electrolyte’?
10. Please provide the comparison table, which you need to prove that your material is superior to previously reported literature.The authors should add some literature descriptions to make the manuscript more convincing. I would like to suggest the authors cite the following relevant articles to enhance the background, supporting the importance of SCs; “Nanomaterials, 2022, 12(22), 3982”; “Nanomaterials, 2022, 12(18), 2330”; “Nanomaterials, 2022, 12 (14), 3187”.
Author Response
Reviewer 2:
Authors reported “Polypyrrole-coated low-crystallinity iron oxide grown on carbon cloth enabling enhanced electrochemical supercapacitor performance”. The organization of this work is good, and the discussion is well organized. The characterization and calculation are both solid for the conclusion. Nevertheless, I have some comments which are listed below.
- The synthesis scheme is not clear, and it should be revised in a detailed way.
Response:
As suggested, we updated the description of experiment section.
- The calculation formula of specific capacity ( mF cm-2) is not accepted by researchers. Specific capacity and specific capacitance are two concepts. The author shall unify the units as F g-1 or mA h g-1.
Response:
We appreciated for the review’s good comments. As suggested, we carefully reviewed the manuscript, and changed the “specific capacity” to “specific capacitance” in line 283. As for the self-support electrode with binder free feature, area specific capacitance is usually employed to evaluate the charge storage. Due to the feature of the direct formation on the current collector surface, the comparison of area specific capacitance is more intuitive. As suggested, here, we also provided the mass loading of active materials Fe2O3, D-Fe2O3, and D-Fe2O3@PPy in experiment section. The related mass specific capacitances (F g-1) were also shown accordingly.
- The author claimed that their work is a novel investigation, however, myriads of works related to “ D-Fe2O3@PPy “ have been published to date. So, authors should change the way of the presentation focusing on novelty. The introduction should be improved with a paragraph describing the novelty and importance of the work.
Response:
We appreciated for the review’s good comments. Fe2O3 as the psedocapacitive material has attracted much attention. Some Fe2O3/Polypyrrole nanocomposites as the electrode for ion batteries and supercapacitors has been reported so far. In this work, we conducted NaBH4 treatment to provide reduction reaction for preparing the low-crystallinity and high active metal oxide electrode. As excepted, D-Fe2O3 delivered high capacitance but poor cycling stability (Figure 5f). Then, we used electrodeposition method to construct Fe2O3/PPy layer. It shows that the construction of low-crystallinity Fe2O3 coated by PPy layer can simultaneously boost capacitance and electrochemical stability. The design of low-crystallinity metal oxide and polymer nanocomposite is expected to be widely applicable for the development of the state-of-the-art electrode. As suggested, we added the related description in introduction (Please see line 65–71, line 77–84).
- The authors must carefully claim their novelty in the INTRODUCTION. In addition, the authors need to do some formatting errors that should be carefully checked and corrected in the text.
Response:
As suggested, we added the detailed discussion in introduction section (Please see line 65–71, line 77–84).
- The source and purity of all chemicals used should be specified.
Response:
As suggested, the source and purity of all chemicals and substrate was added in 2.1 material section (Please see line 100–105).
- The surface area measurements are very important for active electrodes in supercapacitor applications. Please provide nitrogen adsorption and desorption (BET analysis) of the active materials (D-Fe2O3@PPy).
Response:
We appreciate for the review’s good suggestion. The specific surface area is significant to estimate the D-Fe2O3@PPy/CC electrode. We decided to measure the specific surface area by the N2 adsorption-desorption experiment. But the self-support electrode is not suitable for BET measurement due to the considerable low quality of active materials, which is different from powder samples, because the BET test needs at least 200 mg active material for accurate measurement. Electrochemical active surface area (ECSA) is also the important factor for catalysts with nanostructures. The D-Fe2O3@PPy/CC electrode were cycled in the non-faradaic regions under the same condition. However, we found the faradic reaction cannot be ignored during the CV test, which is due to the redox reaction of Fe2O3 plays a key role. As shown in Figure 5e, D-Fe2O3@PPy/CC electrode exhibited a higher capacitive contribution compared with pristine Fe2O3/CC. The enhanced capacitive contribution indicates a larger active surface area. We hope this measurement can also demonstrate the possible large surface area of D-Fe2O3@PPy/CC to a certain extent.
- The authors have missed their mass loadings of “1.) MnS; 2.) Ni(OH)2, and 3.) MnS@Ni(OH)2” electrode materials should be included in the revised manuscript.
Response:
As suggested, we gave the mass loading of active materials Fe2O3, D-Fe2O3, and D-Fe2O3@PPy in experiment section. The related mass specific capacitances (F g-1) were also given accordingly (Please see line 117, 126, and 138).
- The selected area electron diffraction (SAED) is very crucial in supercapacitor applications. Please provide SAED of the active materials (D-Fe2O3@PPy).
Response:
As suggested, SAED pattern was given in Figure S3. The related discussion was added (Please see 248–251).
- The authors should confirm that in the “Experimental section” in the Electrochemical studies having a “ 1 M Na2SO4” electrolyte solution. Here, my point is why the author not used ‘‘Li2SO4, SPVA-HRG, 1 M H2SO4’ and ‘other sources’ instead of ‘Na2SO4’ electrolyte’?
Response:
In the electrochemical evolution system of Fe2O3 electrode, KOH and Na2SO4 were considered to be the best electrolyte, because their low-cost feature compared with Li+ aqueous electrolyte. The acid electrolyte is usually used in carbon-based electrode material, but can etch the metal oxide especially Fe2O3. Our previous work demonstrated the Na+ insertion process in Fe2O3 nanoparticles. The diffusion-control contribution could be improved by using Na2SO4 solution as electrolyte (ACS Appl. Mater. Interfaces 2021, 13, 45670–45678.). In addition, neutral electrolytes exhibit the feature of safety and environmentally friendly. Hence, we chose the 1 M Na2SO4 as the electrolyte for evolution of electrodes.
- Please provide the comparison table, which you need to prove that your material is superior to previously reported literature.The authors should add some literature descriptions to make the manuscript more convincing. I would like to suggest the authors cite the following relevant articles to enhance the background, supporting the importance of SCs; “Nanomaterials, 2022, 12(22), 3982”; “Nanomaterials, 2022, 12(18), 2330”; “Nanomaterials, 2022, 12 (14), 3187”.
Response:
We have added the related references (Ref. 30, 14, and 15). The comparison table was shown in Table S1.
